# Design of Constructed Wetland Treatment Measures for Highway Runoff in a Water Source Protection Area

**Guoping Qian, Chang Wang, Xiangbing Gong * , Hongyu Zhou and Jun Cai**

National Engineering Laboratory for Highway Maintenance Technology, Changsha University of Science and Technology, Changsha 410114, China; guopingqian@sina.com (G.Q.); wangchang@stu.csust.edu.cn (C.W.); zhouhongyu23@126.com (H.Z.); fromhscaijun@163.com (J.C.)

* Correspondence: xbgong@csust.edu.cn; Tel.: +86-176-0365-8650

**Abstract:** Road runoff contains high levels of pollutants, such as heavy metals and hydrocarbons. If they are directly discharged into sensitive water bodies, they will cause irreversible pollution and damage to the water environment. Furthermore, the leakage of hazardous chemicals into sensitive waters will lead to serious consequences, so determining how to deal with road surface runoff has become an urgent problem. This research adopts a scheme for collecting and processing road runoff in a water source protection area using artificial wetlands. After optimizing and improving the general vertical flow of the wetland structure, a composite wetland structure and a relatively novel tandem wetland structure are proposed. An indoor model is established for experiments on various main wetland structure schemes. The results show that the two newly proposed wetland structures improve the possibility of water level control in general vertical flow structures. At the same time, the movement distance of the water flow in the wetland structure is changed to improve the treatment effect of runoff. The removal effect of composite and tandem wetland structures for heavy metals, petroleum substances, and COD (chemical oxygen demand) is significantly better than that of general vertical flow structures. Among them, the composite structure is better than the tandem structure at removing heavy metals, petroleum substances, and COD. However, due to the water discharge method of the structures, the latter has a better effect than the former in the treatment of suspended substances.

**Keywords:** constructed wetland; road runoff; water source protection area; wetland structure; substrate filler

## 1. Introduction

In recent years, with the increase in the mileage of highway construction, an increasing number of researchers at home and abroad have begun to pay attention to the impact of highway construction on the surrounding natural environment. In highway construction and operation, road runoff carrying pollutants into the local water body will have a certain negative impact on the local environment. Pollutants in road surface runoff come from the scouring action of rainwater, which mainly includes solid suspended substances (SSs), heavy metals (Cu, Zn, Pb, etc.), nutrients, oils, and some other organic matter [1–4]. To study these problems in depth, and then to solve them, researchers have studied the characteristics of pollutants in runoff. Some researchers highlighted that RDS (road-deposited sediments) pollution and first flush on runoff are the main two sources of pollutants in road runoff [5]. In road-deposited sediments, most of the heavy metals are attached to particles, the size of which is less than 0.15 mm. Among them, the concentration of Cu and Pb in fine particles is relatively high, while zinc (Zn) has a higher concentration in the coarse and fine particles [6,7]. Some experiments collected runoff samples from overpass sections, university districts, residential areas, and sidewalks to obtain the concentration of heavy metals and study the outflow laws of heavy metals in the path flow [8]. Kim et al. [9]

measured the pollutants in the bridge runoff of Gongju during seven rainfall events and found that the average concentrations of COD (chemical oxygen demand) and SS in each rainfall event varied significantly, ranging from 45.48 to 198.95 mg/L and from 24.73 to 305.34 mg/L. The concentrations of PAHs (polycyclic aromatic hydrocarbons) and heavy metals in asphalt pavement runoff were measured by Gjessing et al. [10], and it was found that the concentrations of Pb, Zn, TOC (total organic carbon), and COD in asphalt pavement runoff were 3~5 times higher than those in cement concrete pavement. Especially when the road passes through the water source protection area, it will have a bigger influence on the local natural environment if the road surface runoff is not purified. Therefore, there is an urgent need to purify the road runoff before discharging it.

At present, the most commonly used measures for the treatment of road runoff worldwide are vegetation control, retention ponds, constructed wetlands, infiltration systems, etc. The SWMM (storm water management model) of the US Environmental Protection Agency was used to simulate the infiltration of plant filter belt runoff, and the results show that plant filter belts have a strong diluting effect on the pollutants in the road-stream [11]. Some researchers have compared the pollutant concentration of road runoff after treatment with plant filter belts and those without treatment. The results show that plant filter belts can effectively remove particulate pollutants (including Pb, Zn, and PAH) in runoff [12]. Vegetation filters (VFSs) and grass (GS) are proven to be highly effective in removing total suspended solids (TSS) [13]. In the study of the relationship between the purification effect of grass planting and the runoff speed, Deletic et al. [14] found that the runoff speed is inversely proportional to the effect of removing suspended solids. Vegetation control is a good method for controlling road runoff owing to its wide applicability, along with its convenient design and construction, but it is easy to cause the accumulation and blockage of pollutants using this method.

The detention tank is divided into the dry detention tank and wet detention tank to remove pollutants. Some researchers are studying the treatment effect of the coagulant in the sedimentation tank on road runoff. Research shows that the total removal rate of particles and metals in the coagulation process reaches 90%, and the dissolution of Cr, Cu, and Pb reaches 40% [15]. Cheng et al. [16] proposed and studied a first-flush capture and detention tank to receive rainwater runoff from asphalt pavement, and the results showed that the equipment has a 90% TSS removal rate. Studies by R. et al. [17] show that both dry and wet retention ponds are effective in removing heavy metals and other contaminants. Comings et al. [18] showed that the removal rates of SS, Pb, and Zn in wet retention ponds were 70%, 70%, and 40%, respectively. A study by Swedish researcher Lundberg et al. [19] found that, as the treatment time increased, the removal of COD from the retention ponds was almost zero and other pollutants could not be removed.

The infiltration system is an effective way to deal with road surface runoff by filtering and intercepting. The main forms are a seepage well, an open seepage pit, a seepage ditch, a porous road surface, and a multi-stage infiltration system [20]. Research by Wu et al. [21] highlighted that most of the runoff pollutants in vegetation depressions are removed owing to osmosis, and the average event mean concentrations of total suspended solids (TSS), total nitrogen (TN), and total phosphorous (TP) can be reduced by 53%, 67%, and 25%, respectively. Other studies have shown that the porous pavement structure can effectively reduce the concentration of pollutants in runoff [22]. Pagotto et al. [23] showed that the removal rate of SS, heavy metal Pb, and hydrocarbons was up to 85%, 78%, and 92%, respectively, in a porous pavement structure. However, the porous pavement is more prone to blockage, and the pavement often needs to be replaced during maintenance. The maintenance cost is high, so the applicability of the pavement structure is poor. Infiltration systems are generally more suitable for areas with low groundwater levels. Due to the many factors affecting treatment efficiency, the current application range is temporarily small.

In the process of road runoff purification, the constructed wetland has the advantages of a large hydraulic treatment load, a large variety of pollutants being removed, a good comprehensive purification effect, and coordination with the ecological environment.

Through comparative experiments, Zhou et al. [24] found that constructed wetlands can effectively remove heavy metal elements in road runoff. Through tracking and monitoring, Gill et al. [25] found that the removal rate of heavy metal elements Cd, Cu, Pb, and Zn in runoff from wetland was much higher than the theoretical calculation, and the wetland could effectively remove heavy metal elements in runoff. Choi et al. [26] designed and proposed a hybrid constructed wetland to treat road runoff. The monitoring results show that the removal efficiency of total suspended solids (TSS), COD, total nitrogen (TN), total phosphorous (TP), and heavy metals in the mixed constructed wetland is at least 60%, which is 0–10% higher than that of a single constructed wetland. Senduran et al. [27] designed a pocket wetland to treat runoff in the Lake Sapanca catchment area, and the average removal efficiency for the total suspended solids (TSS), total nitrogen (TN), total phosphorous (TP), Cu, and Zn in the runoff was 52%, 26%, 63%, 7%, and 55%, respectively. Jinhui et al. [28] designed a horizontal subsurface flow constructed wetland using adsorption medium/substrate, and through water quality testing, the average removal efficiency of total suspended solids (TSS), COD, total Kjeldahl nitrogen, NH4+-N, and total phosphorous (TP) was 86.5%, 68.1%, 78.25%, 95.2%, and 64.85%, respectively. Terzaki et al. [29] monitored the treatment of road runoff for two years by constructing wetlands using the surface flow and underground flow. The results showed that the two types of wetland structures have excellent removal effects on COD, total suspended solids (TSS), total nitrogen (TN), total phosphorous (TP), Cu, Ni, Pb, and Zn. Research facts show that the constructed wetland can effectively remove various pollutants in road runoff, and also has the advantages of low construction cost and long service life. Furthermore, it has a certain landscape effect and can beautify the road environment [30].

However, currently constructed wetlands exhibit problems such as difficult water level control, slow runoff treatment rates, and the possibility of the treated sewage causing secondary pollution. Therefore, this paper analyzes the structural characteristics of the currently commonly used constructed wetlands, proposes two new wetland structure optimization schemes on the basis of the existing general vertical subsurface wetland structure, and creates an indoor model for experiments testing the runoff purification effects.

## 2. Materials and Methods

### 2.1. Design of Constructed Wetland

2.1.1. Structure Types of Constructed Wetland

According to the flow model of the water body, a constructed wetland can be divided into three basic types: a surface flow wetland, a horizontal subsurface flow wetland, and a vertical subsurface flow wetland. The components of each wetland mainly include matrix fillers (such as soil, sand, and gravel), aquatic plants, and various microorganisms. After runoff flows into various wetlands, it is subjected to comprehensive purification treatment by precipitation, filtration, adsorption, ion exchange, plant absorption, and microbial degradation [31].

2.1.2. Structural Designs of Constructed Wetland

Considering the high requirement of water quality in the water source protection area, to improve the treatment effect of runoff as much as possible and to reduce the amount of land required, the type of wetland is designed as a subsurface flow constructed wetland. This experiment mainly designs three kinds of wetland structures, which are the general vertical flow wetland structure, the compound subsurface flow wetland structure, and the serial subsurface flow wetland structure. The specific structures are shown in Figure 1 below.

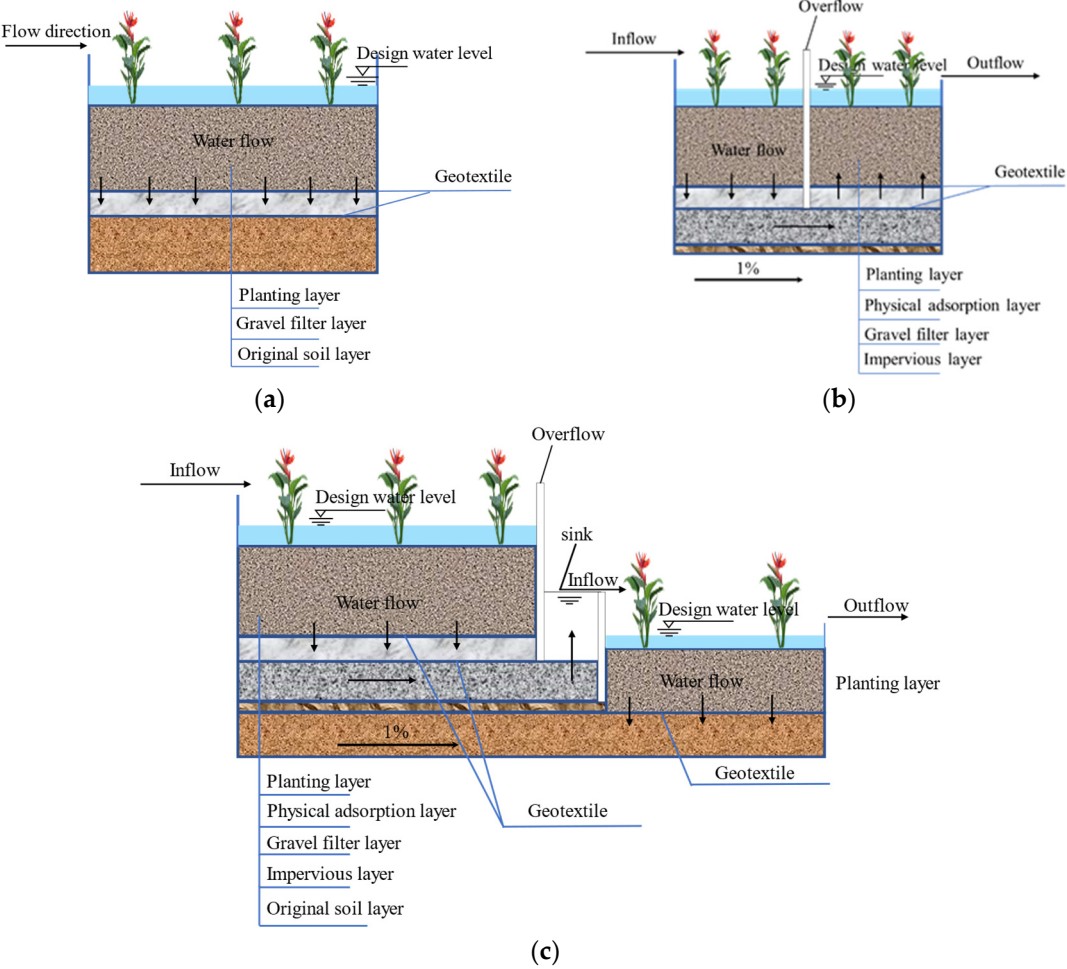

**Figure 1.** The constructed wetland structure designed in this experiment: (**a**) general vertical flow wetland structure, (**b**) compound constructed wetland structure, and (**c**) tandem subsurface flow wetland structure.

In the vertical flow wetland structure (as shown in Figure 1a), large particles of suspended solids from runoff are deposited on the surface of the planting layer; then, heavy metal ions and petroleum pollutants will be absorbed by plant roots and degraded by soil microorganisms. Finally, they will be added to the groundwater through the original soil via the filter layer. This structure may lead to rainwater infiltration from the top of the wetland to overflow over time, thus affecting the purification effect. Therefore, based on the principle of the communicating device, we designed the wetland as a composite constructed wetland structure that combines vertical and horizontal subsurface flow (as shown in Figure 1b). This structural design causes the height of the runoff inlet, concrete partition wall, and outlet to gradually decrease, so that when the runoff is too high, the excess water can overflow through the top of the concrete wall into the wetland on the other side, solving the problems existing in the structure of the general vertical flow wetland. However, due to the increase in water movement distance, the purification process will be prolonged, and the runoff treatment rate will be reduced.

The hydraulic characteristics of constructed wetlands, such as water level control and water movement distance, also affect the purification efficiency of wetlands [32,33]. Therefore, we proposed a tandem subsurface flow wetland structure (as shown in Figure 1c). The structure uses a water trough to connect two vertical subsurface wetlands in a series, which improves the runoff treatment rate by reducing the movement distance of the water. This construction has the advantages of water level control, purification effect, and runoff

processing rate, yielding a strong practical performance. Therefore, in this paper, a novel wetland structure scheme is proposed in the design of the constructed wetland.

### 2.2. Laboratory Test Model Creation

The test model is made of a stainless steel plate, according to the structural scheme of a general vertical flow wetland, composite subsurface flow wetland, and tandem subsurface flow wetland, as shown in Figure 2. (The unit in the figure is cm).

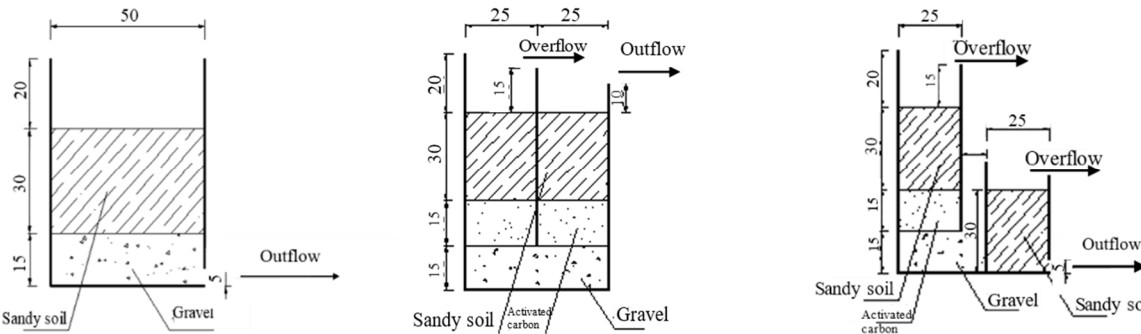

**Figure 2.** Model section size diagram (unit: cm).

The matrix material in the planting layer was sandy soil, mixed with 50% medium and coarse sand, and 50% natural soil; reeds and cattails were selected as the plants for the planting layer; in the filter layer, the gravel had a particle size of 8~16 mm. The physical adsorption layer in the composite structure and the tandem structure used activated carbon as the adsorption material. The thickness of the implant layer was 30 cm, the physical adsorption layer was 15 cm, and the filter layer was 15 cm. The cleaned matrix materials were filled into the model according to the order of matrix layers.

The experiment was divided into three groups: general vertical flow wetland structure, compound constructed wetland structure, and tandem subsurface flow wetland structure; parallel experiments were performed using three identical indoor models for each structure, and the purification effects of the three wetland scheme structures on the pollutants were tested, respectively.

### 2.3. Preparation of Road Runoff Water

The road runoff water used for the test was prepared by dissolving the sediments on the road and mixing them uniformly in tap water. Due to the influence of runoff and wind, most of the road sediments were distributed on both sides of the road, within a width of about 50 cm from the curb. Therefore, the road surface sediments cleaned in this experiment came from the 50 cm width of the road surface on the left side of the road at Wanjiali South Road, Yuhua District, Changsha City, China, which were mainly dust. After cleaning, a filter was used to remove large particles, such as leaves. The substance was then sealed and stored.

### 2.4. Detection Indicator

The indicators for water quality testing included SS, heavy metals (Pb, Zn, and Cu), COD, and petroleum. By detecting the concentration of each indicator in the initial water sample and the purified water sample, the removal effect was studied.

The testing methods and main testing equipment of each index are shown in Table 1.

**Table 1.** Road runoff detection indicators and detection equipment.

| Serial Number | Detection Indicator | Referenced Standards | Testing Equipment |
|---|---|---|---|
| 1 | SS | GB 11901-89 [34] | Oven, balance |
| 2 | Zn | GB 7475-87 [35] | GDYS-201M multi-parameter water quality analyzer |
| 3 | Pb | GB 7475-87 | GDYS-201M multi-parameter water quality analyzer |
| 4 | Cu | GB 7475-87 | GDYS-201M multi-parameter water quality analyzer |
| 5 | COD | HJ 828-2017 [36] | GDYS-201M multi-parameter water quality analyzer |
| 6 | Petro | SL 93.2-94 | MAI-50G infrared oil meter |

*2.5. Test Setup*

2.5.1. Determination of Test Time

In order to determine when the test stabilized the removal rate of pollutants in the runoff, three sets of runoff treatment tests were carried out using a series structure model, and the test times were 60 min, 120 min, and 180 min, respectively. Then, we calculated the removal rate of Zn, Pb, and Cu at each time in the 60 min, 120 min, and 180 min test groups, respectively. The removal ($C_s$) rate was calculated as follows:

$$C_S = \frac{C_0 - C_i}{C_0} \times 100\% \tag{1}$$

where $C_0$ denotes the concentration of pollutants in the original water sample, and $C_i$ denotes the concentration at each time point.

The test results are shown in Figure 3:

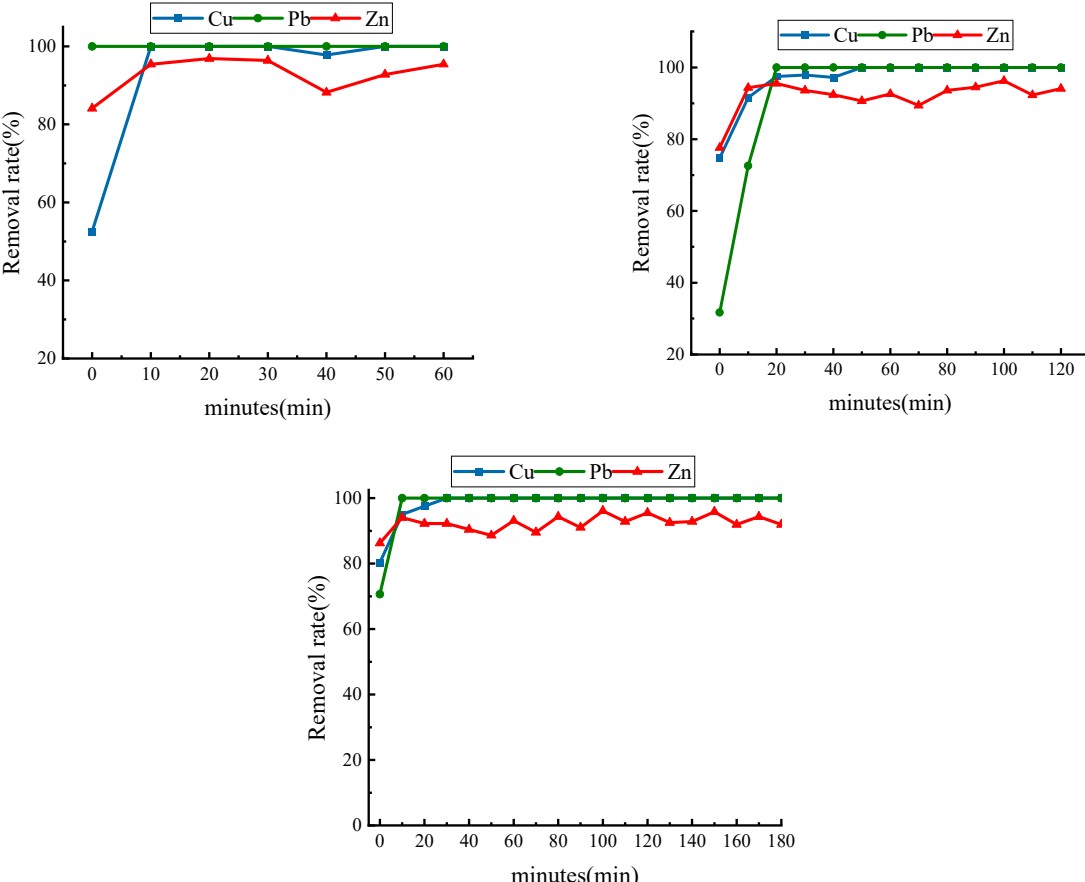

**Figure 3.** Comparison chart of 180 min, 120 min, and 60 min curves.

It can be seen from the figure that the changes in the removal rate of pollutants in the three groups of tests stabilized after 30 min. This showed that the treatment of runoff in this test could reach stability within 60 min, so each group of tests adopted 60 min as the test time.

### 2.5.2. Experimental Procedure

In order to obtain the raw water sample required for the test model, we dissolved 2.5 kg of the sediment in approximately 160 L of water and stirred it. The sewage was then added to the model, and the time from the start of water addition to the outflow was recorded. When the water flowed out of the water outlet of the model, water samples were collected every 10 min for a total of 60 min, and were numbered in sequence of 0 min, 10 min, 20 min, 30 min, 40 min, 50 min, and 60 min.

### 3. Results and Discussion

The test results of the pollutant concentration in each group of experiments are summarized in Figure 4. For the convenience of description, in Figure 4, the general vertical flow wetland structure, compound structure, and tandem structure are abbreviated as GVFS, CS, and TS, respectively.

In Figure 4, the concentration of SS and COD was higher at 0 min, and the concentration decreased significantly after 10 min of treatment in the constructed wetland, which was particularly obvious in the composite structure and tandem structure wetlands. The initial concentrations of heavy metals Pb, Zn, and Cu were not high, and the concentrations decreased significantly after wetland treatment. In particular, after 60 min of treatment with the two newly designed constructed wetland schemes, the concentrations of Pb and Cu in the effluent samples were almost 0 mg/L.

To more intuitively understand the treatment effects of different structures on pollutants, the pollutant removal rate at each time from 0 min to 60 min and the average removal rate within 60 min were calculated. The calculation results are shown in Figures 5 and 6.

From the analysis of Figures 5 and 6, we can see:

The pollutant removal rate of each group of structures showed a similar change over time. The removal rate increased in the first 30 min, and then fluctuated within a certain range. This indicates that the structure of each scheme can effectively purify road runoff, and the removal rate of various pollutants will gradually stabilize in the short term.

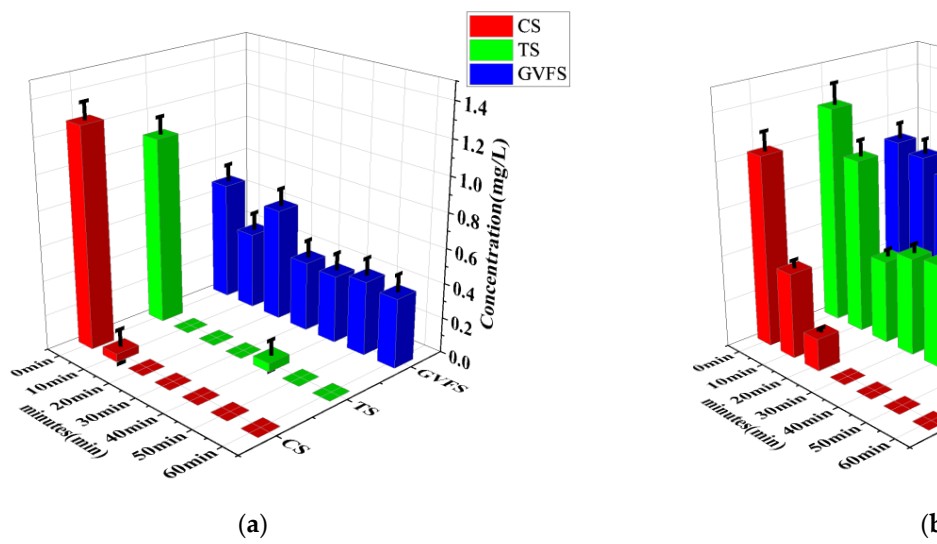

(**a**)

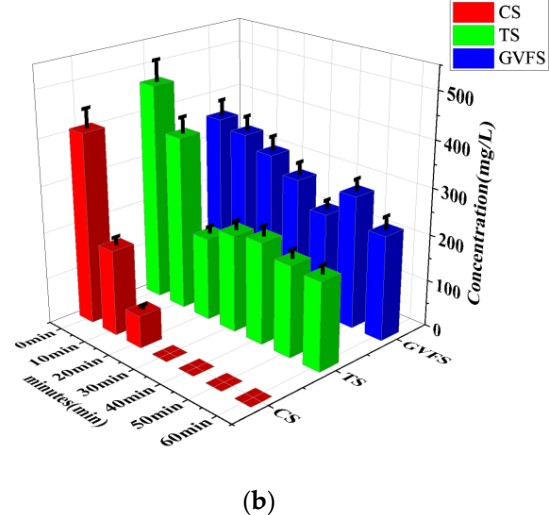

(**b**)

**Figure 4.** *Cont.*

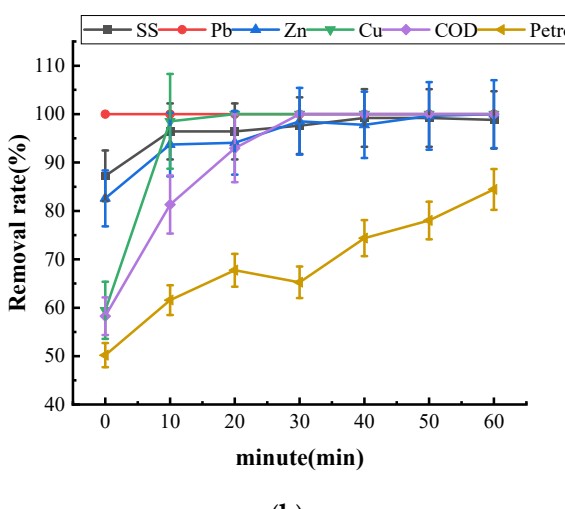

**Figure 4.** The concentration of different pollutants varies over time in different wetland structures. (**a**) Cu; (**b**) COD; (**c**) SS; (**d**) Pb; (**e**) Petro; (**f**) Zn.

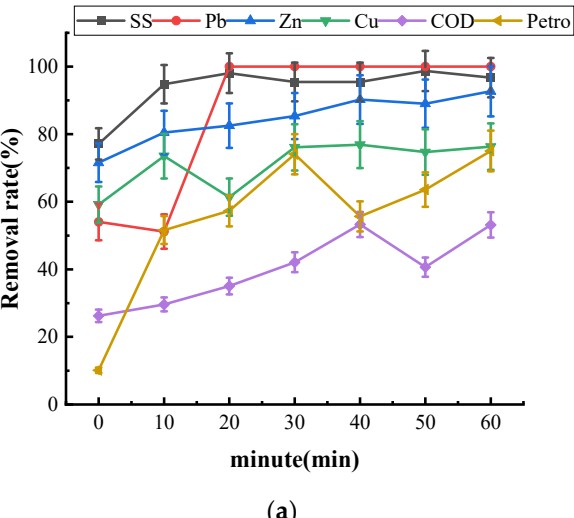

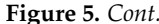

**Figure 5.** *Cont.*

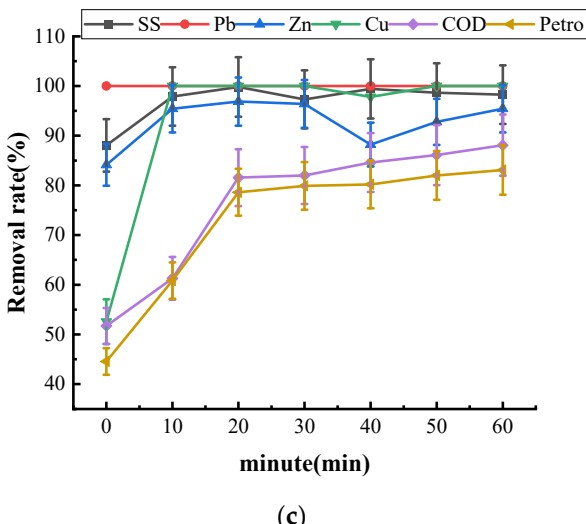

(**c**)

**Figure 5.** Pollutant removal rate of each structure: (**a**) general vertical flow wetland structure, (**b**) compound constructed wetland structure, and (**c**) tandem subsurface flow wetland structure.

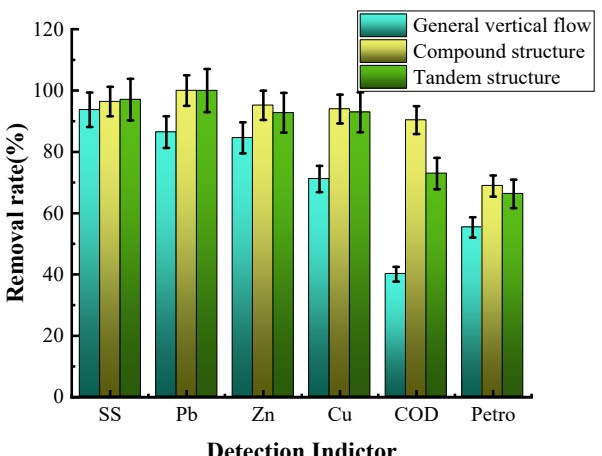

**Figure 6.** The average removal rate of pollutants in 60 min.

The three structural schemes had good treatment effects on suspended solids, with removal rates of 93.74%, 96.41%, and 97.05%, respectively. The removal rates of Zn were 84.55%, 95.19%, and 92.75%; the removal rates of Cu were 71.14%, 94.00%, and 92.91%; the removal rates of Pb were 86.46%, 100%, and 100%; the removal rates of petroleum were 55.35%, 68.81% and 66.30%; and the removal rates of COD were 40.01%, 90.36%, and 72.93%.

The composite and tandem wetland structures were significantly better than the general vertical flow structure for the removal of various pollutants. The reason for this is that after the runoff enters from the left, it undergoes physical filtration, plant root absorption, and microbial degradation after the first purification. Then, it penetrates vertically into the bottom of the first-level wetland and into the gravel filter layer. Finally, the runoff flows from the filter layer up to the second-level wetland and out of the wetland after the second purification. In the whole process, the runoff undergoes two purification processes. It is worth noting that, in the treatment of suspended matter, since the composite structure will drain upwards and the water flow will disturb the sandy soil, the treatment effectiveness is not as high as for the tandem structure.

In this study, the two novel constructed wetland structures that we proposed had a good purification effect on pollutants in road runoff. In our actual engineering investigation, we found that constructed wetlands could also be used as aquatic habitats in urban areas,

as shown in Figure 7. This wetland is located in the South Second Ring Road of Guangzhou. It is mainly used to collect and purify the road surface at the toll station and the bridge deck runoff on the ramp bridge. There are few pollutants in the urban road surface runoff, so the artificial wetland can be used as an aquatic habitat to maintain the urban ecological balance.

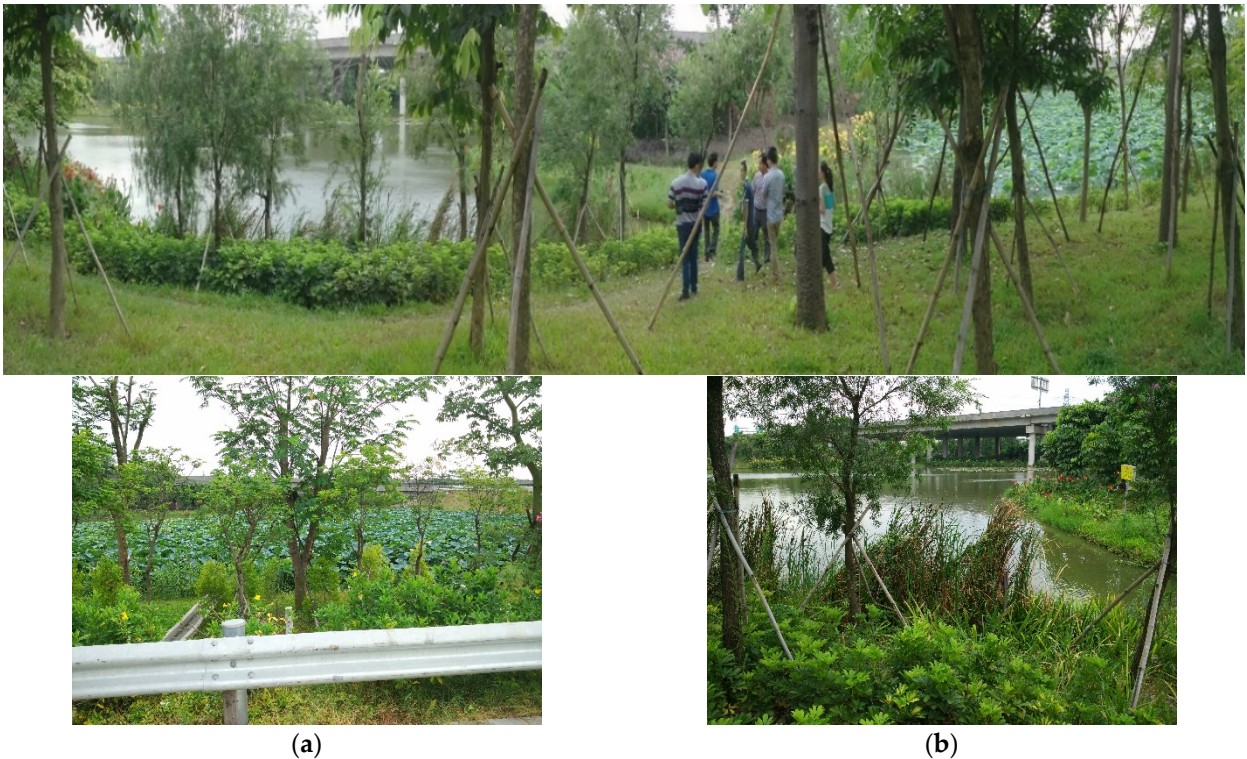

(**a**)                                                                                                  (**b**)

**Figure 7.** Guangzhou South Second Ring Constructed Wetland. (**a**) First-level wetland; (**b**) Second-level wetland.

The wetland uses two surface-flow artificial wetlands connected by a natural ditch to implement the graded treatment of runoff. The first-level wetland is used for the preliminary purification of runoff, and lotus flowers are planted in the wetland. When the water level in the first-level wetland is full, it flows into the second-level wetland through natural ditches and is further filtered here. Plants such as reeds are planted in the second-level wetland. Plants in wetlands grow well and attract many animals to settle there, so it is possible to apply artificial wetlands to aquatic habitats. When constructed wetlands are degraded from water treatment functions to ecosystems, they are especially likely to be used as animal habitats to maintain biodiversity [37].

## 4. Conclusions

This paper combines the advantages and disadvantages of typical structures for the treatment of road runoff and proposes a scheme for collecting and processing road runoff in the water source protection area using artificial wetlands. Two schemes for optimizing the wetland structure were proposed. To simulate the purification effect of the three wetland structures in the treatment of road runoff, an indoor test model was constructed. The results show:

(1) Based on the general vertical subsurface flow wetland, two optimization schemes for the wetland structure are proposed. They are a U-shaped composite subsurface flow wetland, combining vertical and horizontal subsurface flow, and a series subsurface flow wetland structure, with two vertical subsurface flow wetlands in a series. The structure improves the shortcomings of the general vertical flow wetland's water level control

difficulties and the treatment effect of runoff by changing the flow distance of the water flow in the matrix filler.

(2) The decontamination effect of the general vertical flow, composite underflow, and tandem underflow wetland structures were compared by testing the quality of influent and effluent water. The removal effect of the new structures is obviously better than that of the general vertical flow structure. The new structures are similar in the removal rate of heavy metals and petroleum substances, and the removal rate of each heavy metal index is above 90%. In particular, the removal rate of Pb was close to 100%. The removal rate of petroleum substances was slightly worse, but still reached 68.81% and 66.30%, respectively. The removal rate of COD was quite different, 90.36% and 72.93%, respectively. The compound type is better than the series type in terms of removal efficiency.

(3) In general, the removal effect of the composite wetland structure is the best, followed by the tandem type, but the former is slightly worse than the latter in terms of the removal effect of suspended matter. The reason for this phenomenon may be related to the water outlet method of the structure. The structure uses upward drainage, and the water flow disturbs the sandy soil material, which causes the turbidity of the effluent body to increase.

**Author Contributions:** Conceptualization, G.Q. and X.G.; methodology, G.Q. and X.G.; investigation, C.W.; validation, C.W., X.G. and J.C.; supervision, H.Z. and J.C.; writing—original draft preparation, C.W. All authors have read and agreed to the published version of the manuscript.

**Funding:** The research was supported by the National Natural Science Foundation of China (grant no.: 52078065) and the science and technology innovation Program of Hunan Province (grant no.: 2020RC4048).

**Institutional Review Board Statement:** Not applicable.

**Informed Consent Statement:** Not applicable.

**Acknowledgments:** This study was completed at the School of Traffic and Transportation Engineering, Changsha University of Science and Technology.

**Conflicts of Interest:** The authors declare no conflict of interest.

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
