# Peer review of "Design of Constructed Wetland Treatment Measures for Highway Runoff in a Water Source Protection Area"

_sustainability, doi:10.3390/su14105951_

Round 1

Reviewer 1 Report

The important and actual subject of highway runoff treatment in water source protection area is taken up in the study. I would like to underline that the manuscript is carefully prepared, but I strongly recommend to improve English style and nomenclature. Moreover, I have noticed some following shortcomings:

  • Lines 121÷124: It should be written whether the structures are adapted (existing), or they are the new Authors’ proposition, or which structure  is existing and which is new
  • Lines 127 and 135: The plural should be used (types and designs)
  • Figure 1(b): The arrows on the right side of the cross-section should point upwards due to the presence of an impermeable layer at the bottom
  • Line 160: “Figure. 1(a)” should be replaced by “Figure 1(b)”
  • Lines 166÷172: There should be a reference to Figure 1(c)
  • Figure 2: “inflow” should be replaced by “outflow”
  • Lines 187: It is not clear – 3 groups need a short description
  • Line 257÷258: The formula (1) should go to the section 2. Material and Methods

Author Response

Thanks for your comment and it helps us to improve the quality of this paper. The details of the revisions to the manuscript and responses to your comments are in the cover letter. Please see the "Response to Reviewer 1 Comments" in cover letter.

Reviewer 2 Report

There is an excessive use of word "scholar(s)" in your paper, try to avoid this as much as possible.

The other comments/observations, in the attached file.

Author Response

Thanks for your comment and it helps us to improve the quality of this paper. The details of the revisions to the manuscript and responses to your comments are in the cover letter. Please see the "Response to Reviewer 2 Comments" in cover letter.

Reviewer 3 Report

Dear Authors

I have read your manuscript with great interest. There are, however, some points in the text that need improvement. I have listed my recommendations below.

1) The acronym COD appears twice in the abstract - please write it out in full;

2) There are many acronyms throughout the manuscript, and they make reading difficult and unintelligible. I emphatically recommend authors to eliminate those acronyms from the text, or to reduce their use to what is strictly necessary.

3) There are acronyms that are not explained textually (eg PAH and U.S. EPA SWMM);

4) What plant species are used in the planting layer? How are the pollutants retained by those plants eliminated?

5) The researchers demonstrated that the tandem structure is more effective for the retention and removal of pollutants. But what about its cost? Is the tandem structure more expensive than the other models tested by the scientists? If it is more expensive, why should decision-makers adopt it?

6) I am an urban ecologist and I visited China a few years ago. The image of Chinese cities full of bicycles that I expected to find belongs to the past. Vehicle traffic - private automobiles - is increasing rapidly and I recommend to the authors, in the Discussion, that they write a few paragraphs about the possibility of using those wetlands in tandems for the protection of aquatic habitats in urban areas.

Author Response

Thanks for your comment and it helps us to improve the quality of this paper. The details of the revisions to the manuscript and responses to your comments are in the cover letter. Please see the "Response to Reviewer 3 Comments" in cover letter.

Round 2

Reviewer 2 Report

After complying to the reviewers' comments, the authors rendered this paper publishable as is.